# Heat Stress during Summer Attenuates Expression of the Hypothalamic Kisspeptin, an Upstream Regulator of the Hypothalamic–Pituitary–Gonadal Axis, in Domestic Sows

**DOI:** 10.3390/ani12212967

**Published:** 2022-10-28

**Authors:** Hwan-Deuk Kim, Young-Jong Kim, Min Jang, Seul-Gi Bae, Sung-Ho Yun, Mi-Ree Lee, Yong-Ryul Seo, Jae-Keun Cho, Seung-Joon Kim, Won-Jae Lee

**Affiliations:** 1College of Veterinary Medicine, Kyungpook National University, Daegu 41566, Korea; 2Department of Veterinary Research, Daegu Metropolitan City Institute of Health & Environment, Daegu 42183, Korea; 3Institute of Equine Medicine, Kyungpook National University, Daegu 41566, Korea

**Keywords:** kisspeptin, domestic sows, heat stress, summer infertility, c-Fos, hypothalamic–pituitary–gonadal axis

## Abstract

**Simple Summary:**

The present study compared the expression of hypothalamic kisspeptin, an upstream regulator of the hypothalamic–pituitary–gonadal axis, in different seasons (summer and spring) to assess the effect of heat stress on reproductive performance in domestic sows. Sows in summer showed a decreased pregnancy rate and litter size and increased secretion of stress-related hormones than those in spring. Heat stress in summer lowered hypothalamic kisspeptin expression and reduced the activity of kisspeptin neurons. Due to the lowered expression of hypothalamic kisspeptin, the downstream reproductive hormones and the number of large ovarian follicles were reduced in summer. These findings may have significant implications in the fields of reproductive biology and the livestock industry for heat stress control.

**Abstract:**

The release of reproductive hormones in the hypothalamic–pituitary–gonadal (HPG) axis is regulated by its upstream regulator, kisspeptin, and influenced by external stresses, including heat stress. Since the effect of heat stress (summer infertility) on hypothalamic kisspeptin expression in domestic sows is not yet understood, the present study attempted to identify changes in kisspeptin expression in different seasons (summer and spring). The high atmospheric temperature in summer decreased the pregnancy rate and litter size and increased stress-related hormones as a chronic stressor to domestic sows. The hypothalamic kisspeptin expression in summer was decreased regardless of the estrus phase and negatively correlated with atmospheric temperature, indicating that high temperature decreased kisspeptin. When the activity of hypothalamic kisspeptin neurons in the follicular phase was assessed using c-Fos staining, a decreased number of kisspeptin neurons coexpressing c-Fos was observed in domestic sows in summer. Accordingly, lower expression of kisspeptin induced decreased levels of HPG axis-related reproductive hormones, such as gonadotropins and estrogen, and fewer large ovarian follicles. In conclusion, the present study demonstrated that reduced kisspeptin expression and its neuronal activity in the hypothalamus under heat stress in summer induced downregulation of the HPG axis and caused summer infertility in domestic sows.

## 1. Introduction

Pigs (*Sus scrofa*) are among the most agriculturally important livestock in the animal industry. However, several factors, such as imbalanced nutrition, infectious diseases, genetics, husbandry, and stress control, directly or indirectly influence the reproductive performance of livestock [1]. Therefore, the management of sows is the key to improving the reproductive performance of pig herds as well as an economic activity for profitability; reproductive failure, such as abnormal cyclicity and delayed puberty, is the biggest reason for culling gilts and sows [2,3].

The release of reproductive hormones in the hypothalamic–pituitary–gonadal (HPG) axis to control the reproductive performance and puberty of female livestock is governed by complex networks with stimulatory or inhibitory signals (positive or negative feedback), which are dependent on two distinct phases: the follicular and luteal phases [4]. Gonadotropin-releasing hormone (GnRH) is secreted from the hypothalamus and transported through the hypophyseal portal veins into the anterior pituitary gland, which is followed by the release of gonadotropins (GTHs), such as follicle-stimulating hormone (FSH) and luteinizing hormone (LH). They are responsible for the development of ovarian follicles, where estradiol (E2) or progesterone (P4) is released from large and preovulatory follicles during the follicular phase or corpus luteum (CL) during the luteal phase, respectively [5,6]. During the past several decades, kisspeptin neurons have been shown to control the HPG axis as an upstream regulator that plays a critical role in controlling the onset of puberty, GnRH and GTH secretion, ovarian cyclicity, and fertility [4,7]. There are species-specific differences in the location of kisspeptin neurons in the hypothalamus; however, they are mainly localized in the hypothalamic region of the arcuate nucleus (ARC), medial preoptic area (mPOA), and periventricular nucleus (PeN) and placed adjacent to GnRH neurons. In pigs, both ARC and PeN are known to be kisspeptin neuron-localized regions as GnRH pulse and surge mode generators, respectively [8,9].

Stress in livestock is defined as a perceived threat by several stimuli to the homeostasis of the body, such as noise, transportation, food deprivation, weaning, social isolation, physical restraint, and heat [1,4]. It can be divided into acute stress for short-term negative situations and chronic stress for a long-lasting stressed condition, and the former activates the sympathetic-adrenal-medullary (SAM) axis; however, the latter stimulates hypothalamic-pituitary-adrenocortical (HPA) axis. The hormones in the HPA axis consist of corticotropin-releasing hormone (CRH), adrenocorticotropic hormone (ACTH), and glucocorticoid (cortisol) to control the physiological balance between anabolism and catabolism under stressed conditions [1,10]. In particular, it has been reported that stress hormones from the HPA axis impair reproductive performance in females in terms of an alteration of synthesis/secretion of GTHs, change in downstream reproductive hormones, abnormal ovarian cyclicity, weakening ovarian follicle growth, and infertility [4,10,11].

Meanwhile, an air temperature range of 20–22 °C in adults has generally been suggested as a suitable temperature for raising pig herds [12,13]. However, the physical features of pigs, such as thick subcutaneous fat and a small number of sweat glands, make it difficult to cope with high atmospheric temperatures during summer, which is considered an unavoidable chronic stressor in farm pigs and induces summer infertility [14]. The annual economic loss due to heat stress was calculated as $299 million in the pig industry in the USA in 2003; summer infertility affects 5–25% of breeding females [7,15]. Pigs with summer infertility present several reduced reproductive performances with respect to an increased number of sows returning to estrus after mating, weak or undetectable estrus signs, delayed puberty in gilts, the extended interval of weaning to estrus in sows, increased anestrus period, higher rates of pregnancy failure, and reduced litter size [7,14].

Therefore, a further study to understand heat stress in pigs is required to prevent the loss of productivity and establish a proper strategy to improve reduced reproductive performance. Although it has been known for many decades that stimulation of the HPA axis by stress inhibits the activity of the HPG axis, the effect of summer heat stress on kisspeptin expression in the hypothalamus has not yet been identified. Therefore, the present study attempted to identify whether kisspeptin expression in domestic sows was affected by different temperatures in other seasons, especially spring and summer. Understanding how high temperatures in summer influence kisspeptin expression may be important for the management of farm animals and the handling of summer infertility in sows.

## 2. Materials and Methods

### 2.1. Ethics Statement

All animal specimen sampling procedures were approved by the Institutional Animal Care and Use Committee at Kyungpook National University (approval number: 2021-0098).

### 2.2. Chemicals and Media

All chemicals and reagents were obtained from Thermo Fisher Scientific (Waltham, MA, USA), unless otherwise specified.

### 2.3. Acquisition of Samples from Domestic Sows in Different Seasons

Healthy sows at a mooring in the abbatoir were selected by two veterinarians under basic physical examination, including normal body condition, the absence of external wounds, and non-diarrhea. Additionally, the absence of infectious disease was examined in the dressed carcasses of sows after slaughter. The sampling was conducted in different seasons including spring (SP; maximal atmospheric temperature: 18.4–21.1 °C) and summer (S; maximal atmospheric temperature: 29.5–33.4 °C) for five days each; these seasons were selected in accordance with variations in accompanying air temperatures in Korea, which could represent suitable temperatures for pigs (SP) or heat stress-inducing temperatures causing summer infertility (S) [12,13,14]. The domestic sows (total *n* = 50; *n* in SP = 25 and *n* in S = 25) were three-way crossbred by Landrace×Yorkshire×Duroc (LYD), approximately 2-year-old, weighing approximately 200 kg, non-pregnant, experienced for 3.4 parities, fed for fully balanced commercial feeds, free to access water ad libitum, and came from similar breeding-sized domestic pig farms (*n* = 5) located near Daegu-si, Republic of Korea (latitude: 128.63° E; longitude: 35.87° N) and not equipped with air temperature controller; the number of pig farms (*n* = 5) in the present study was considered as sufficient to minimize farm-to-farm variations due to different breeding environments. We further investigated each farm’s reproductive performance records regarding pregnancy rate (%) and the number of littermates per delivery in different seasons (SP and S) to determine whether summer infertility arose in response to high temperatures during summer in these farms. Before slaughter, the dermal temperature on the neck of the sows in a mooring was measured using a non-contact thermometer (HFS-1000, HuBDIC Co., Ltd., Anyang, Gyeonggi-do, Korea) in triplicate. In addition, fresh whole blood samples were collected via jugular venipuncture in ethylenediaminetetraacetic acid tubes (BD Falcon, Becton, NY, USA), and the supernatant was immediately isolated by centrifugation at 1000× *g* for 15 min at 4 °C. The isolated serum for hormone assays was stored in a deep freezer at −80 °C until use (total *n* = 50; *n* in SP = 25 and *n* in S = 25). All reproductive organs, including the hypothalamus and ovaries from domestic sows, were collected during slaughter at a local abattoir. After opening the skull (total *n* = 50; *n* in SP = 25 and *n* in S = 25) upon slaughtering, the whole brain including the hypothalamus was fixed with 4% paraformaldehyde (Duksan Chemical, Ansan, Korea) for immunofluorescence assays (total *n* = 18; *n* in SP = 9 and *n* in S = 9) or sagittally dissected to bluntly isolate hypothalamus by micro forceps for Western blotting (total *n* = 32; *n* in SP = 16 and *n* in S = 16; Appendix A). Both ovaries were immediately fixed with 4% paraformaldehyde (Duksan Chemical) for classification of the estrus cycle and differential counting of ovarian follicle sizes in the follicular phase (total *n* = 50; *n* in SP = 25 and *n* in S = 25). During ovary sampling, the subjects who showed cystic follicles or corpus hemorrhagicum in the ovary were intensively excluded owing to reproductive hormone imbalance or intermediate characteristics between the follicular and luteal phases, respectively (Appendix A).

### 2.4. Classification of Estrus Cycle in Sows

As hormonal and ovarian events could present the status of the estrus cycle, we classified the status of the estrus cycle of each subject as follicular or luteal (total *n* = 50; *n* in SP = 25 and *n* in S = 25). By morphological observation of the ovarian surface on gross and cross-sectioned ovaries, the follicular phase (*n* in SP = 17 and *n* in S = 17; Appendix A) or luteal phase (*n* in SP = 8 and *n* in S = 8; Appendix A) was determined when growing follicles from small/medium follicles to Graafian follicles with regression of CL from previous cycles, or mature CL formation with growing small follicles but not large follicles was observed, respectively [16]. Additionally, the levels of E2 and P4 in the serum were further assessed to confirm the estrus cycle of each sow by enzyme-linked immunosorbent assay (ELISA).

### 2.5. ELISA for Reproductive and Stress-Related Hormones of Sows in Different Seasons

Since reproductive and stress-related hormones are considered biomarkers that can inform the status of animals, the levels of several hormones were evaluated in domestic sows during different seasons (SP and S; total *n* = 50; *n* in SP = 25 and *n* in S = 25) in accordance with the manufacturer’s instructions and a previously published article [1,10]. ELISA kits were obtained from Cayman Chemical Company (Ann Arbor, MI, USA) for FSH, LH, E2, P4, and corticosterone or MyBioSource, Inc. (San Diego, CA, USA) for kisspeptin, GnRH, and cortisol. Briefly, after the stored sera were thawed, a mixture of sample, enzyme-immunoassay buffer, tracer, and antiserum (FSH, LH, E2, P4, and corticosterone) was incubated for 60–90 min at room temperature (RT; 20–25 °C). Thereafter, samples in the 96-well plate were reacted with Ellman’s reagent for 15–60 min in an incubator. In the case of others (kisspeptin, GnRH, and cortisol), samples and antibodies were added to antigen-precoated 96 well, and the wells were incubated for 40–60 min at 37 °C. Thereafter, the wells were incubated with horseradish peroxidase (HRP)-conjugate for 30 min at 37 °C and reacted with 3,3′,5,5′-tetramethylbenzidine substrate for 20 min at 37 °C. The reacted 96-well plates were read at a wavelength of 405–450 nm using a microplate reader (Epoch, Biotek, Winooski, VT, USA). The concentration of each hormone in the serum was calculated using a four-parameter logistic fit with free software (www.myassay.com; accessed on 5 November 2021).

### 2.6. Western Blotting for Reproductive and Stress-Related Hormones of Sows in Different Seasons

Western blotting for kisspeptin, GnRH, and CRH expression in the hypothalamus under different estrus cycles and seasons was conducted as previously described (total *n* = 32; *n* in SP = 16 and *n* in S = 16) [4,10]. Briefly, the snap-frozen hypothalami were homogenized, lysed with radioimmunoprecipitation assay buffer supplemented with a proteinase inhibitor, and centrifuged at 13,000 × *g* for 5 min at 4 °C. The supernatants were gently isolated and quantified for the total amount of protein in accordance with a protocol using a bicinchoninic acid protein assay reagent kit. Equal amounts of total protein (25 μg) from each sample were separated using 10% sodium dodecyl sulfate-polyacrylamide gel electrophoresis, followed by transfer onto polyvinylidene difluoride membranes (Millipore, MA, USA). Thereafter, the membranes were blocked with 3% bovine serum albumin (BSA) for 60 min at RT and incubated with rabbit polyclonal anti-kisspeptin antibody (Abcam, Cambridge, UK; 1:250 dilution with 1% BSA), mouse polyclonal anti-GnRH (MyBioSource, Inc.; 1:1000 dilution with 1% BSA), rabbit polyclonal anti-CRH antibody (Proteintech, Inc., Rosemont, IL, USA; 1:500 dilution with 1% BSA), and mouse monoclonal anti-glyceraldehyde 3-phosphate dehydrogenase (GAPDH; Abcam; 1:2000 dilution with 1% BSA) primary antibodies overnight at 4 °C. The membranes were incubated with an HRP-conjugated goat anti-mouse or anti-rabbit IgG (1:3000 dilution with Tris buffer saline containing 0.1% tween-20) for 60 min at RT and developed on radiographic films (AGFA, Mortsel, Belgium) using an enhanced chemiluminescence kit. ImageJ software (National Institutes of Health, Bethesda, MD, USA) was used to quantify the intensities of the bands. The levels of kisspeptin, GnRH, and CRH expressions were normalized to that of GAPDH.

### 2.7. Immunofluorescence Assay to Determine c-Fos Co-Expressing Kisspeptin Neurons of Sows in Different Seasons

The c-Fos has been identified as a marker of neuronal activity because it is expressed in depolarized neurons [17]. In particular, the co-expressing pattern of target proteins with c-Fos has been widely applied to study active hypothalamic neurons in various species [17,18,19]. One-third of kisspeptin neurons express c-Fos at the time of the GnRH and LH surge at PeN near the third ventricle (3V) of the brain [18]. Therefore, to identify the alteration between kisspeptin neurons and their neuronal activity depending on the season, an immunofluorescence assay with kisspeptin and c-Fos in the follicular phase was conducted (total *n* = 18; *n* in SP = 9 and *n* in S = 9). The fixed hypothalami of sows in the follicular phase in different seasons were isolated from whole fixed brains, processed, dehydrated through a graded ethanol series (70–100%), embedded in paraffin, and sectioned at a thickness of 5 mm routinely. The slides were then deparaffinized, rehydrated through graded ethanol series and distilled water, treated with 0.01 M citrate buffer (pH 6.0) at 95 °C for 30 min, cooled at RT for 60 min, treated with 3% H_2_O_2_ for 30 min, washed with phosphate-buffered saline with 0.1% triton-X, blocked with 2% normal goat serum (NGS; Vector Laboratories, Newark, CA, USA) for 60 min at RT, and incubated with a rabbit polyclonal anti-Kisspeptin antibody (1:50 dilution with 2% NGS) together with a mouse polyclonal anti-c-Fos antibody (Santa Cruz Biotechnology, Inc., Dallas, TX, USA; 1:100 dilution with 2% NGS) at 4 °C overnight. The slides were then washed with PBS, incubated with goat anti-rabbit IgG conjugated with Alexa Fluor 488 (1:1000 dilution with PBS; green fluorescent protein, GFP) together with goat anti-mouse conjugated with Alexa Fluor 594 (1:1000 dilution with PBS; red fluorescent protein, RFP) at RT for 60 min, counterstained with 4′,6-diamidino-2-phenylindole for 5 min, and mounted with VectaShield (Vector Laboratories). Since it was addressed that kisspeptin-positive cells were mainly distributed in PeN near 3V in pigs during the late follicular phase, these regions were randomly selected (*n* = 3 per slide) and observed in a fluorescent microscope at proper wavelengths; the images were taken under same fluorescent exposure time. The percentage (%) of kisspeptin neurons co-expressing c-Fos was then compared following different seasons [9].

### 2.8. Classification of Ovarian Follicles of Sows in Different Seasons

The development of ovarian follicles during the follicular phase is directly influenced by several reproductive hormones, including GnRH, FSH, and LH. In pigs, a general consensus is that ovarian follicles can be classified as small (0.20–0.30 cm), medium (0.31–0.39 cm), large (0.40–1.00 cm), and cystic [2,16]. According to these criteria, the ovarian follicles of the sows in the follicular phase (total *n* = 34; *n* in SP = 17 and *n* in S = 17) in response to different seasons were counted (%).

### 2.9. Statistical Analysis

All assays were performed in triplicate. The number of samples to each assay was arranged in Appendix A. The values between two groups (SP and S) or more than three groups (follicular phase in SP, luteal phase in SP, follicular phase in S, and luteal phase in S) were statistically analyzed using Student’s t-test or one-way ANOVA with Tukey’s post hoc test using SPSS 12.0 (SPSS Inc., Chicago, IL, USA), respectively. Additionally, Pearson’s correlation analysis was conducted between several factors (dermal temperature, atmospheric temperature, and/or various hormone levels). Differences were considered statistically significant at *p* < 0.05. The graphs are presented as mean ± standard error (SEM).

## 3. Results

### 3.1. Reproductive Performance during Summer Infertility in Domestic Sows

We first investigated the heat stress factors (atmospheric and dermal temperature) and reproductive performance in the farms where domestic sows in the present study were raised, to characterize the current state of summer infertility in subjects. As expected, a significantly (*p* < 0.05) higher atmospheric and dermal temperature were observed in the summer group (Figure 1A,B), and both temperature values presented a positive correlation as linear regression (*p* = 0.00; r = 0.582) (Figure 1C), which indicated that high atmospheric temperatures in summer affected body temperature in domestic sows. In terms of reproductive performance, the summer group exhibited a significant (*p* < 0.05) decrease in pregnancy rate (%) and the number of littermates per delivery relative to the spring group (Figure 1D,E). Overall, these data implied that the reproductive performance of domestic sows in the present study was affected by high temperatures in summer causing summer infertility.

### 3.2. Assessment of Stress during Summer Infertility of Domestic Sows

We further assessed whether high summer temperatures could affect stress-related hormones in the study subjects. The expression of CRH, the upstream stress hormone in the HPA axis, was significantly (*p* < 0.05) increased in the hypothalamus during the Western blotting assays (Figure 2A,B) in the summer group, relative to the spring group. Additionally, as expected, significant (*p* < 0.05) elevations in downstream stress hormones (corticosterone and cortisol) were identified in the summer group by ELISA (Figure 2C,D), and the highly positive correlations between downstream stress hormones and atmospheric temperature were assessed (Figure 2E,F; r = 0.683 or 0.703 in corticosterone and cortisol, respectively). These results indicate that the domestic sows in the present study suffered from heat stress during summer, which is possibly followed by summer infertility (Figure 1).

### 3.3. Expression of the Hypothalamic Kisspeptin and Its Downstream Hormone (GnRH) in Domestic Sows in Different Seasons

Since the high temperature in summer caused a reduction in reproductive performance as well as induction of the increased activity of the HPA axis, we further analyzed alterations in hypothalamic kisspeptin expression as an upstream regulator of the HPG axis in different seasons and its effect on GnRH expression in the hypothalamus as a direct downstream reproductive hormone of kisspeptin in domestic sows. Western blotting (Figure 3A) demonstrated significantly (*p* < 0.05) lower kisspeptin expression in the hypothalamus regardless of the estrus cycle in the summer groups (Figure 3B). Additionally, GnRH expression was consistent with the hypothalamic kisspeptin expression (Figure 3C). In the ELISA assay for kisspeptin and GnRH concentration in the serum, similar to the results of the Western blotting assay, a significant (*p* < 0.05) reduction in both hormones could be found in the summer group, relative to the spring group (Figure 3D,E). Notably, kisspeptin concentration in the serum presented a highly negative correlation with atmospheric temperature (*p* = 0.00; r = −0.771), indicating that higher temperature directly decreased kisspeptin secretion (Figure 3F). Additionally, the reduced kisspeptin concentration in the serum was positively correlated with a decrease in its downstream hormone, GnRH (*p* = 0.00; r = 0.629), which indicated that lowered kisspeptin expression owing to high temperature negatively affected the HPG axis (Figure 3G).

### 3.4. Activity of Kisspeptin Neurons in the Follicular Phase in Different Seasons

As the expression of kisspeptin was reduced under heat stress conditions in summer, further analysis to assess their neuronal activity was conducted by fluorescent immunohistochemical staining to detect kisspeptin and c-Fos (a marker for active neurons)-positive cells in the region of PeN near 3V of the hypothalamus in the follicular phase of domestic sows. There was no significance but kisspeptin-positive cells (GFP-positive cells) were more abundantly localized at PeN near 3V of the hypothalamus in the spring group, implying that the hypothalamic kisspeptin expression at the follicular phase in domestic sows was decreasingly affected by high temperatures during summer (Figure 4A–C). Interestingly, the population of kisspeptin neurons co-expressing c-Fos (GFP and RFP double-positive cells), i.e., active kisspeptin neurons, was significantly (*p* < 0.05) reduced in the summer group relative to the spring group, indicating that heat stress during summer affected not only kisspeptin expression but also the activity of kisspeptin neurons in the hypothalamus of domestic pigs (Figure 4D–F).

### 3.5. Effects of Reduced Kisspeptin Expression on HPG Axis

We mainly focused on whether hypothalamic kisspeptin expression was affected by summer infertility, and it was demonstrated that high temperature during summer induced low expression of kisspeptin and activity of kisspeptin neurons, which was followed by a decrease in GnRH expression. As kisspeptin is an upstream regulator of the HPG axis, we additionally investigated the functional alteration of HPG axis-related hormones and organs with regard to FSH/LH secretion in the pituitary gland, and E2/P4 secretion and folliculogenesis in the ovary. Both FSH and LH secretions from the pituitary gland were significantly (*p* < 0.05) reduced during the follicular phase, possibly due to a decrease in kisspeptin and GnRH levels (Figure 5A,B). In the ovary, while P4 levels during the luteal phase were not observed between different seasons, E2 levels during the follicular phase were significantly (*p* < 0.05) reduced in the summer group relative to those in the spring group (Figure 5C,D). The results showing a significant (*p* < 0.05) increase in the number of small follicles (immature) and a decrease in the number of large follicles (mature) in the summer group compared to the spring group could be derived from the reduced levels of FSH and LH and were associated with a lower level of E2 (Figure 5E). Collectively, the reduction in hypothalamic kisspeptin expression under heat stress in summer induced less secretion of GnRH, FSH, and LH, which was followed by a lower population of mature follicles in the ovary and a lower concentration of E2.

## 4. Discussion

Fluctuating climates, such as noise, cold, heat, humidity, rain, and wind, can be chronic stressors in livestock and affect the endocrine system. Among these, heat stress during summer in the Northern Hemisphere has been extensively studied [7]. Animals under heat stress present changes in behavior (long time spent standing, searching for cooler areas, high water intake, and low feed intake); physiological status (sweating, panting, and increased body temperature and respiration); and secretion of several hormones [1]. Therefore, breeding management against these stressors is of great importance to the animal industry because they can inhibit the balance of reproductive hormones, resulting in decreased reproductive performance and consequent economic loss [8]. In pigs, the management of sows is the key to improving the reproductive performance of pig herds, as well as an economic activity for profitability. However, the yield of reproductive performance is affected by chronic stress, including heat, resulting in summer infertility, delayed puberty, the extended interval from weaning to estrus, higher rates of pregnancy failure, and reduced litter size in sows [3,7,14]. Therefore, understanding the effect of heat stress in summer on the HPG axis, including an upstream regulator, kisspeptin, is important for establishing a strategy to manage farm animals and handle summer infertility in sows. The present study demonstrated that reduced kisspeptin expression in the hypothalamus and decreased kisspeptin neuronal activity under heat stress in summer induced downregulation of the HPG axis with regard to decreased levels of GnRH, FSH, and LH, which might be followed by reduced folliculogenesis in the ovaries, pregnancy rate (%), and the number of littermates per delivery in sows. To the best of our knowledge, the present study is the first to report a relationship between hypothalamic kisspeptin and heat stress during summer in domestic sows.

Regulation of the HPG axis by kisspeptin expression has been widely investigated in several species. The hormonal systems by which kisspeptin regulates GnRH release in the hypothalamus were evident, showing that kisspeptin induced GnRH secretion in the hypophyseal portal vein in ewes and ovariectomized goats after its intravenous injection [20,21]. Similar results were found in rodents after intracerebroventricular or peripheral administration of kisspeptin [22]. Kisspeptin stimulates the release of GTHs, especially LH, in a GnRH-dependent manner [7]. An increase in LH levels was accompanied by GnRH release following kisspeptin application in ewes [23]. In addition, ewes treated with GnRH-neutralizing antibodies or induced disconnection between the hypothalamus and pituitary gland presented failure of kisspeptin to increase LH levels [24,25]. In pigs, kisspeptin injection into the brain’s lateral ventricles produces surge-like secretion of LH, which is sustained for several hours [5]. In addition, gilts can induce a serum LH increase after kisspeptin injection [26]. Porcine anterior pituitary cells release LH after kisspeptin treatment [27]. In accordance with several articles, the main localized region of kisspeptin is species-dependent and can be consolidated as the ARC and mPOA in ruminants (sheep, cattle, and goat), ARC and anteroventral periventricular nucleus (AVPV) in rodents, and distinctively at ARC and PeN in pigs [8,9,28,29]. The function and role of kisspeptin neurons in the PeN of pigs is similar to that of murine AVPV, which is related to the generation of an estrogen-induced ovulatory surge (GnRH surge) during the follicular phase and increased kisspeptin expression in the PeN [9]. The present study demonstrated that the hypothalamic kisspeptin expression and their neuronal activity at the region of PeN near 3V in the follicular phase were lower in the summer group than in the spring group, indicating that kisspeptin expression was impaired by heat stress during summer (Figure 3 and Figure 4).

Stress can be divided into acute stress for short-term negative situations and chronic stress for long-term stress conditions. The SAM axis is turned on in response to acute stress in the animal to produce norepinephrine in the peripheral sympathetic nerves and epinephrine in the adrenal medulla, which increases the heart rate, respiration rate, and blood flow [1,30]. However, as the animal can no longer cope with acute stress, the HPA axis is activated to release CRH, ACTH, and glucocorticoids (cortisol) to regulate stress [1,31]. These stress-related hormones regulate processes such as glucose storage and release, immune function, digestion, and reproduction against stressors [11]. However, when the HPA axis is incapable of controlling chronic stress to maintain homeostasis in the body, several disorders and diseases may occur in animals [1]. As reproduction requires a large amount of energy in the body, it is thought that reproductive performance is inhibited by stressors to maintain homeostasis against stressors and to first save energy [32]. Regarding reproductive disorders due to stress, it has been known that stress-related hormones inhibit the HPG axis, especially the GnRH pulse generator, consequently releasing LH and FSH in the pituitary gland [33,34]. In several experimental animals such as ewes, rats, and primates, various stressors including prolonged intermittent foot shock stimulation, confinement, lipopolysaccharide (LPS) injection, administration of CRH or cortisol, social isolation, application of blindfold, and the presence of predator suppressed GnRH secretion and the preovulatory LH surge [11,31,32,33]. Additionally, cortisol application in ovariectomized ewes reduced GnRH responsiveness and sensitivity of gonadotroph cells in the pituitary gland [11,35,36]. In the present study, domestic sows under heat stress during summer (chronic stress) presented stimulation of the HPA axis and a significant elevation of stress-related hormones such as CRH, cortisol, and corticosterone than their counterparts in spring (Figure 2). Moreover, the study subjects in summer experienced stress-related physiological changes due to stress hormone-related reduction in kisspeptin (Figure 3), such as decreased reproductive performance, reduced function of the HPG axis, and less folliculogenesis (Figure 1, Figure 3, and Figure 5).

Recently, accumulating evidence has suggested that stressors can affect not only GnRH release but also the activity of kisspeptin neurons [32]. Consistent with GnRH neurons, kisspeptin neurons are also suppressed by stress hormones because they contain type II glucocorticoid receptors [11,37]. Additionally, because kisspeptin neurons in the ARC are distributed together with the CRH receptor, the stress hormone can directly inhibit the function of kisspeptin neurons [38]. Therefore, administration of CRH, corticosterone, cortisol, and LPS in female rodents induces a reduction in kisspeptin expression in the ARC and mPOA, as well as neuronal activity in the hypothalamus [32,39,40]. Moreover, ewes also exhibited a lower percentage of kisspeptin neurons co-expressing c-Fos than controls after 12 h of LPS administration [19]. However, several reports have proposed that stress levels determine whether kisspeptin expression is altered. When moderate or high doses of LPS were individually administered to rats, only high doses of LPS altered kisspeptin expression in the hypothalamus [41]. A study induced stress by several stressors, such as chronic subcutaneous administration of corticosterone and insulin-induced hypoglycemia in rats, and found that the former decreased kisspeptin expression in the mPOA and ARC; however, the latter did not [39]. These results implied that the alteration of kisspeptin expression was dependent on the severity of stressors and that only chronic or severe stress affects the kisspeptin system [32]. In the present study, the high temperature in summer could induce chronic stress in domestic sows (Figure 2), which could decrease the expression of kisspeptin and its neuronal activity (Figure 3 and Figure 4), consequently reducing GnRH, FSH, LH, and folliculogenesis, and occurring summer infertility with lower pregnancy rate (%) and the number of littermates per delivery (Figure 1, Figure 3, and Figure 5).

The activated HPA axis induced an inhibitory effect on not only the brain but also the ovaries due to reduced GTH secretion in female animals. Cortisol administration in the follicular phase suppressed the preovulatory LH surge in ewes [35]. Regarding folliculogenesis with stress, cortisol treatment in ewes arrested follicular development before follicular maturation up to E2-secreting preovulatory follicles, resulting in decreased E2 levels [42]. A study differentially counted the ovarian follicles of weaning sows depending on the season and found that ovaries with small follicles were more abundant in sows weaned in summer-autumn than in winter-spring [2]. In particular, the failure to grow a sufficient population of large follicles in pigs is highly associated with delayed estrus, reduced ovulation rate, and decreased litter size [16]. The present study demonstrated that sows in summer exhibited decreased or increased numbers of large or small ovarian follicles relative to those in spring; this might be caused by reduced kisspeptin expression, which consequently decreases GnRH, FSH, and LH, and induces lower levels of E2 in summer than those in spring in the follicular phase (Figure 5).

## 5. Conclusions

Heat stress in summer is not the sole factor that causes summer infertility in sows with inadequate secretion of GTH, but several external stressors, such as photoperiod, nutritional stress, and immune function, can be linked [7]. Therefore, the potential to induce summer infertility by several factors has been investigated at the level of the HPG axis or its upstream regulators. In the present study, we demonstrated that heat stress was closely related to attenuated kisspeptin expression and neuronal activity in the hypothalamus of domestic sows, resulting in reduced activity of the HPG axis and consequently lower reproductive performance (summer infertility). Therefore, we highlight the importance of thermoregulation, at least for sows, by installing cooling systems, shade, water vaporization, and more frequent checking for signs of estrus return in weaned sows, especially during summer, to minimize the economic loss during summer. Since the reproductive system in animals is greatly orchestrated, we hope that the effect of heat stress during summer on kisspeptin expression in the present study will have significant implications in the fields of reproductive biology and the livestock industry.

## Figures and Tables

**Figure 1 animals-12-02967-f001:**
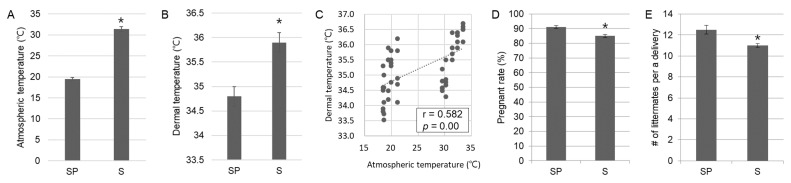
Heat stress factors and reproductive performance during summer of domestic sows. Atmospheric (**A**) and dermal (**B**) temperatures significantly (*p* < 0.05) increased in summer and both are presented as linear regression with a high correlation value (r = 0.582) (**C**). Significant (*p* < 0.05) reductions in reproductive performance such as pregnancy rate (**D**) and the number of littermates per delivery (**E**) are observed in the summer group. The values in B and C are obtained from 50 domestic sows (*n* in SP = 25 and *n* in S = 25). The values in (**D**,**E**) are based on each farm’s reproductive performance records (*n* = 5). Significant differences (*p* < 0.05) are indicated by superscripts * at the top of the bars. Graphs are presented as mean ± SEM. Abbreviations: SP, Spring group; S, Summer group; SEM, standard error of the mean.

**Figure 2 animals-12-02967-f002:**
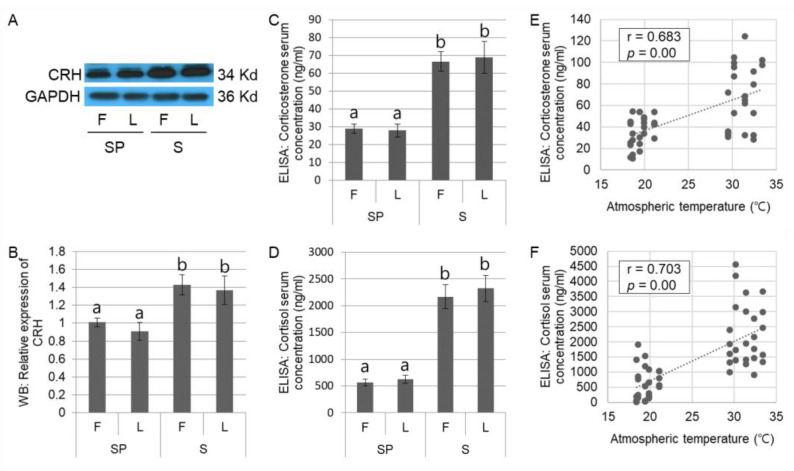
Assessment of stress in the domestic sows during summer. The representative images for CRH and GAPDH in Western blotting assay are presented (**A**). CRH expression is significantly (*p* < 0.05) increased in summer in Western blotting assay (**B**). Elevations of corticosterone (**C**) and cortisol (**D**) in summer are identified using ELISA and positive correlations between these stress hormones and atmospheric temperature are assessed (r = 0.683 in corticosterone and r = 0.703 in cortisol; (**E**,**F**). The Western blotting (**A**,**B**) is conducted using 32 samples (*n* in SP = 16 and *n* in S = 16), and the results by ELISA (**C**–**F**) are obtained from 50 domestic sows (*n* in SP = 25 and *n* in S = 25). Significant differences (*p* < 0.05) among groups are indicated by different letters at the top of the bars. Graphs are presented as mean ± SEM. Abbreviations: CRH, corticotropin-releasing hormone; GAPDH, glyceraldehyde 3-phosphate dehydrogenase; WB, Western blotting; ELISA, enzyme-linked immunoabsorbent assay; F, follicular phase; L, luteal phase.

**Figure 3 animals-12-02967-f003:**
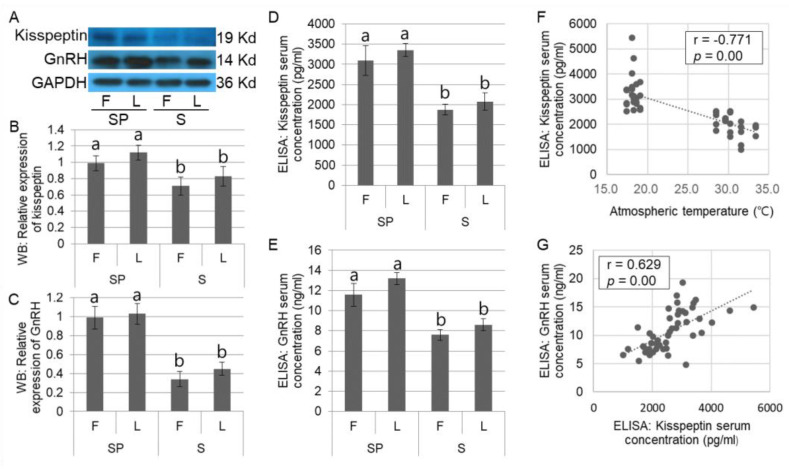
Expression of the hypothalamic kisspeptin and its downstream hormone (GnRH) in domestic sows in different seasons. The representative images for Kisspeptin, GnRH, and GAPDH in Western blotting assay are presented (**A**). The hypothalamic kisspeptin expression was significantly (*p* < 0.05) reduced in the summer group than in the spring group (**B**). GnRH expression is consistent with the results of those of kisspeptin expression (**C**). In the ELISA assay, significant (*p* < 0.05) reductions of kisspeptin and GnRH in the serum were observable in the summer group, relative to the spring group (**D**,**E**). A highly negative correlation between kisspeptin expression and the atmospheric temperature (r = −0.771) or a positive correlation with the reduced kisspeptin and GnRH concentration (r = 0.629) is demonstrated (**F**,**G**). The Western blotting (**A**–**C**) is conducted using 32 samples (*n* in SP = 16 and *n* in S = 16), and the results by ELISA (**D**–**G**) are obtained from 50 domestic sows (*n* in SP = 25 and *n* in S = 25). Different letters at the top of the bars indicate significant differences (*p* < 0.05) among groups. Graphs are presented as mean ± SEM. Abbreviation: GnRH, gonadotropin-releasing hormone.

**Figure 4 animals-12-02967-f004:**
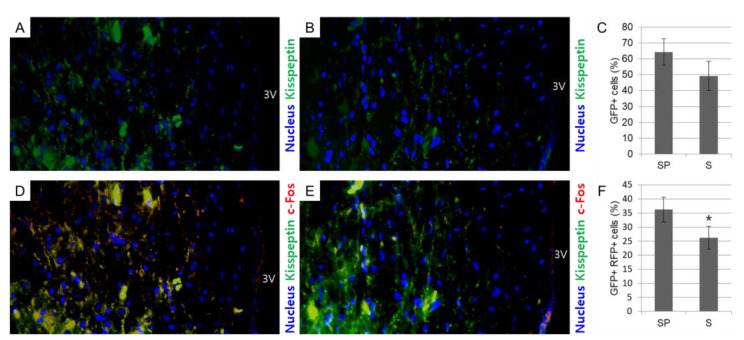
The activity of kisspeptin neurons in the region of PeN near 3V of the hypothalamus in the follicular phase of domestic sows in different seasons. Fluorescent immunohistochemical staining is conducted to identify nucleus (blue), kisspeptin positive cells (GFP), and c-Fos positive cells (RFP) in the region of PeN near 3V. Kisspeptin-positive cells (%) are more abundant in the summer group (**A**–**C**). Kisspeptin neurons co-expressing c-Fos (%) are significantly (*p* < 0.05) reduced in the summer group (**D**–**F**). Significant differences (*p* < 0.05) are indicated by superscripts * at the top of the bars. The fluorescent immunohistochemical staining is conducted using 18 samples (*n* in SP = 9 and *n* in S = 9). Graphs are presented as mean ± SEM. Magnification: ×200. Abbreviations: PeN, periventricular nucleus; 3V, the third ventricle; GFP, green fluorescent protein; RFP, red fluorescent protein.

**Figure 5 animals-12-02967-f005:**
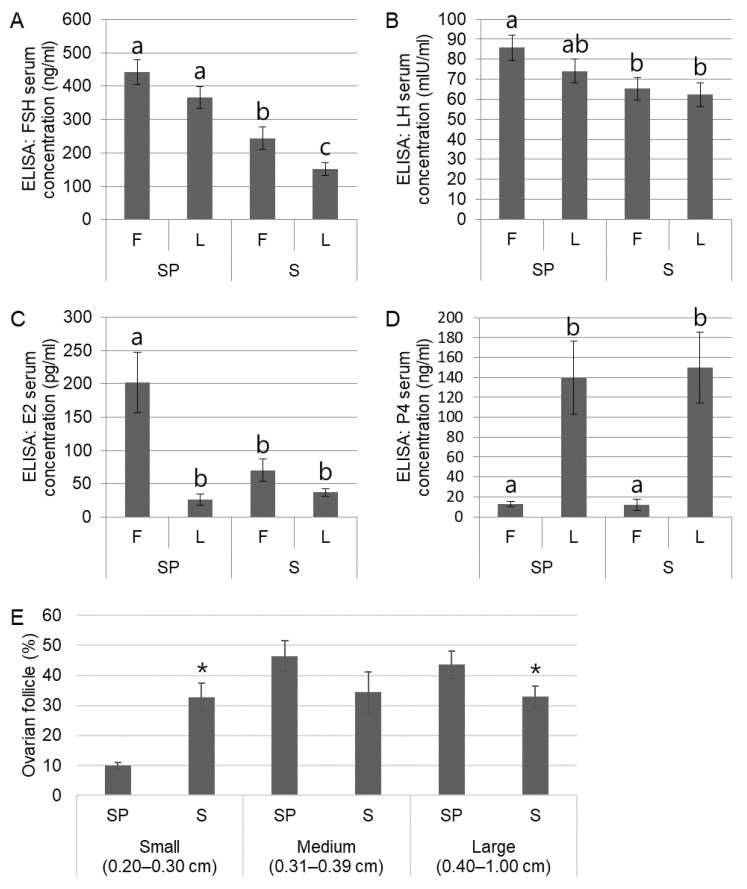
Changes in HPG axis function due to reduced kisspeptin expression. FSH and LH levels from the pituitary gland are significantly (*p* < 0.05) reduced in the summer group (**A**,**B**). Significantly (*p* < 0.05) lower E2 level is observed in the follicular phase of the summer group than in the counterpart (**C**). The pattern of the P4 level is not different between different seasons (**E**). Small or large follicles significantly (*p* < 0.05) increased or decreased in the summer group than those in the spring group, respectively. The results by ELISA (**A**–**D**) are obtained from 50 domestic sows (*n* in SP = 25 and *n* in S = 25). The values for classification of ovarian follicles (**E**) are from 34 animals under follicular phase (*n* in SP = 17 and *n* in S = 17). Significant differences (*p* < 0.05) among groups are indicated by different letters or superscripts * at the top of the bars. Graphs are presented as mean ± SEM. Abbreviations: FSH, follicle-stimulating hormone; LH, luteinizing hormone; E2, estradiol; P4, progesterone.

## Data Availability

All available data are incorporated into the manuscript.

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
