# Peer review of "Heat Stress during Summer Attenuates Expression of the Hypothalamic Kisspeptin, an Upstream Regulator of the Hypothalamic–Pituitary–Gonadal Axis, in Domestic Sows"

_animals, 2022, doi:10.3390/ani12212967_

Round 1
Reviewer 1 Report
1. It is better to involve animals and methods in the abstract.
2. According to the description in the part “2.3. Acquisition of Samples from Domestic Sows in Different Seasons”, the sows involved in this study were from different farms. In addition to the season, other factors such as breeding environment can not be consistent, so how to prove that the differences in results are not caused by other factors? Additionally, how many sows are sampled in each season, and how many sows are sampled in total. More details are needed.
3. In Figure 2A, it is recommended to show the whole image rather than a screenshot. In addition, markers and controls should be included in Figure 2A. The same problems are found in Figure 3A.
4. The number of samples should be clearly described in the Figure captions.
Author Response
We really appreciate to the precious comments and suggestions from reviewer #1, to improve quality of our work. Based on these comments, the manuscript has been carefully revised. A detailed point-by-point reply to the comments of the reviewers (Revision Note) is provided below for your kind perusal, and any revisions made to the manuscript is marked up using the “Track Changes” function in the revised manuscript.
â–ª Comments #1: It is better to involve animals and methods in the abstract.
â–ª Response #1: Due to limitation of the number of words in abstract part of animals (as 200 words), it was not easy to isolate a complete sentence for the detailed information on animals and experimental methods in the abstract part; but we generally described them at the abstract body (e.g. domestic sows, kisspeptin expression, c-Fos staining). We deeply hope that reviewer #1 favorably reconsiders this suggestion.
â–ª Comments #2: According to the description in the part “2.3. Acquisition of Samples from Domestic Sows in Different Seasons”, the sows involved in this study were from different farms. In addition to the season, other factors such as breeding environment can not be consistent, so how to prove that the differences in results are not caused by other factors?
â–ª Response #2: We also agree that other breeding environments such as noise, cold, heat, humidity, rain, wind, photoperiod, and nutritional balance can be chronic stressors, as already written at lane 408-409 and 520-522 in the present manuscript. However, the primary objective of the present study is to uncover the presence of summer infertility (loss of reproductive performance) during summer and relationship between heat stress (summer infertility) and kisspeptin expression, in not the controlled facility but real farms in practice. Therefore, we thought that the results using the samples derived from a farm (n=1) were more unreliable because they could not fully represent the summer infertility; to cover the inconsistency of breeding environment of each farm, we selected the several pig farms (n=5) which might be considered as sufficient to minimize farm-to-farm variations. In addition, when selecting farms, the major criteria were: 1. Placing near Dagu-si, Korea (implying similar atmospheric temperature, humidity, rain, wind, and photoperiod); 2. Similar breeding size (approximately 10,000 pigs); 3. No air temperature control system where the sows could be constantly and directly exposed to high atmospheric temperature in summer. In agreement with reviewer`s comment, the manuscript do not fully contain this background. Therefore, we added some descriptions to highlight this background. The relevant corrections are placed in lane 126 and 128-130.
â–ª Comments #3: Additionally, how many sows are sampled in each season, and how many sows are sampled in total. More details are needed.
â–ª Response #3: In agreement with reviewer`s comment, we also think that the number of samples from different tissues (hypothalamus / ovary / serum) to each assay (western blotting / ELISA / immunofluorescence assay) depending on estrus phase (follicular phase / luteal phase) is better to be described in detail. Therefore, we added the information for the number of samples at each assay as well as prepared the scheme of distribution of samples from different tissues as supplementary figure 2. The relevant corrections are placed in lane 124, 142-143, 145, 147, 148-149, 151, 168-171, 180, 197-198, 227, 253, 256-257, 535-536, and supplementary Figure 2.
â–ª Comments #4: In Figure 2A, it is recommended to show the whole image rather than a screenshot. In addition, markers and controls should be included in Figure 2A. The same problems are found in Figure 3A.
â–ª Response #4: In accordance with 'Instructions for Authors of animals', original images for western blotting should be uploaded together with main manuscript, which might be followed by checking the western blotting procedure by the editorial board; therefore, we already completed to upload the whole western blotting images. In addition, there is a limit to present the whole western blotting images from the total number of 32 samples in a manuscript. Therefore, we prepared the representative images (target proteins: CRH, GnRH, and Kisspeptin; control protein: GAPDH) for western blotting in the manuscript. In agreement with reviewer`s comment, the screenshots may make the readers confused for the procedures of western blotting assays in the present study. Therefore, we revised the relevant figure captions as 'representative images'. The relevant corrections are placed in lane 301-305 and 337-338.
â–ª Comments #5: The number of samples should be clearly described in the Figure captions.
â–ª Response #5: In agreement with reviewer`s comment, describing the detailed number of samples for each assay is better to improve the readability, consistent with response #3. The relevant corrections are placed in lane 286-287, 309-311, 344-347, 371, and 401-403.

Reviewer 2 Report
Comments to the authors
L115: .. and appetite, the absence..
L207: .. was normalized
L218: .. were isolated…
L236: The percentage (%)…
L250: mean±standard deviations
L277: blotting assays
L297: Since the high temperature…..
L322-323: Different letters at the top of the bars indicate significant differences (p<0.05) among groups.
L323-324: Graphs 3 presented as mean±SEM. Abbreviation: GnRH, gonadotropin-releasing hormone.
L380: add appropriate references
L411-412: In pigs, 1 kisspeptin injection into the brain's lateral ventricles produces surge-like secretion of LH, which is sustained for several hours
L441:… foot shock …
Author Response
We really appreciate to the precious comments and suggestions from reviewer #2, to improve quality of our work. Based on these comments, the manuscript has been carefully revised. A detailed point-by-point reply to the comments of the reviewers (Revision Note) is provided below for your kind perusal, and any revisions made to the manuscript is marked up using the “Track Changes” function in the revised manuscript.
â–ª Comments #1: L115: .. and appetite, the absence..
â–ª Response #1: In agreement with reviewer`s comment, we changed the long sentences easily. The relevant correction is placed at lane 115-116.
â–ª Comments #2: L207: .. was normalized
â–ª Response #2: In agreement with reviewer`s comment, we changed the sentence clearly. The relevant correction is placed at lane 216-217.
â–ª Comments #3: L218: .. were isolated…
â–ª Response #3: We deeply apologize a typing error. In agreement with reviewer`s comment, we changed this error. The relevant correction is placed at lane 228
â–ª Comments #4: L236: The percentage (%)…
â–ª Response #4: In agreement with reviewer`s comment, we changed the word properly. The relevant correction is placed at lane 245.
â–ª Comments #5: L250: mean±standard deviations
â–ª Response #5: In accordance with another reviewer`s comment, 'mean±standard deviations' was deleted.
â–ª Comments #6: L277: blotting assays
â–ª Response #6: We deeply apologize a typing error. In agreement with reviewer`s comment, we corrected this error. The relevant correction is placed at lane 295.
â–ª Comments #7: L297: Since the high temperature…..
â–ª Response #7: In agreement with reviewer`s comment, we changed the word properly. The relevant correction is placed at lane 318.
â–ª Comments #8: L322-323: Different letters at the top of the bars indicate significant differences (p<0.05) among groups.
â–ª Response #8: In agreement with reviewer`s comment, we changed the sentence. The relevant correction is placed at lane 346-347.
â–ª Comments #9: L380: add appropriate references
â–ª Response #9: We missed the citation. In accordance with reviewer`s comment, proper reference was added. The relevant correction is placed at lane 410.
â–ª Comments #10: L411-412: In pigs, kisspeptin injection into the brain's lateral ventricles produces surge-like secretion of LH, which is sustained for several hours
â–ª Response #10: In agreement with reviewer`s comment, we changed the sentence. The relevant correction is placed at lane 441.
â–ª Comments #11: L441:… foot shock …
â–ª Response #11: We deeply apologize a typing error. In agreement with reviewer`s comment, we changed this error. The relevant correction is placed at lane 471.

Reviewer 3 Report
This manuscript attempted to identify whether kisspeptin expression in domestic sows was affected by different temperatures in other seasons, especially spring and summer. Understanding how high temperatures in summer influence kisspeptin expression may be important for the management of farm animals and handling of summer infertility in sows.
1. Tittle - adequate
2. Summary - ok
3. Abstract - It is mainly focused on presenting justifications for the study execution. Authors should clearly summarize the objectives, report some details of the methodology (number of individuals per season, methods, biological data collected, etc.) and present some numeric data related to the main findings. Finally, they should clearly present an objective conclusion.
4. Keywords - Authors are advised to avoid indexing terms previously reported at the tittle (maybe also in the abstract).
5. Introduction - At first, authors should correctly write the scientific name of the species Sus scrofa respecting the international rules for this kind of citation. Despite this little mistake, introduction is well written and clearly presents a basis for the study execution. Moreover, objectives are clearly exposed. One point here: despite the interesting design and results, the study is slightly incomplete since it did not present a large evaluation of kisspeptin pattern along all the seasons, and data related to autumn and winter is missing. Anyway, this do not invalidate the study, but remains here as a suggestion for a further evaluation.
6. Methods - I did not understand why the first figure is named as a "supplementary"one; this seems to be unnecessary.
- With regards to the measurement of steroid hormones, was that conducted in an unique moment for the occasion of the slaughtering?Couldn't the slaughter stress influence the hormones profile?
- Please provide more statistical details. Were the data checked for normality and homocedasticity? Did you need to transform any data? What groups were compared using ANOVA followed of Student's t test? Did you check the existence of correlations among what data? Why did you have to represent data using standard error and deviation, and why not using only one representation for all the data?
7. Results - Findings are clear and well presented.
8. Discussions - Results are well-discussed. Sometimes, it seems that authors report a lot of literature review, but it generally helps them to explain their results. Moreover, conclusions are adequate and linked to the main aims of the study.
9. References are ok.
-
Author Response
We really appreciate to the precious comments and suggestions from reviewer #3, to improve quality of our work. Based on these comments, the manuscript has been carefully revised. A detailed point-by-point reply to the comments of the reviewers (Revision Note) is provided below for your kind perusal, and any revisions made to the manuscript is marked up using the “Track Changes” function in the revised manuscript.
â–ª Comments #1: This manuscript attempted to identify whether kisspeptin expression in domestic sows was affected by different temperatures in other seasons, especially spring and summer. Understanding how high temperatures in summer influence kisspeptin expression may be important for the management of farm animals and handling of summer infertility in sows.
â–ª Response #1: We deeply appreciate to the favorable reviews.
â–ª Comments #2: Tittle - adequate
â–ª Response #2: We sincerely appreciate to your positive comment.
â–ª Comments #3: Summary - ok
â–ª Response #3: We deeply appreciate to the favorable comment.
â–ª Comments #4: Abstract - It is mainly focused on presenting justifications for the study execution. Authors should clearly summarize the objectives, report some details of the methodology (number of individuals per season, methods, biological data collected, etc.) and present some numeric data related to the main findings. Finally, they should clearly present an objective conclusion.
â–ª Response #5: Due to limitation of the number of words in abstract part of animals (as 200 words), it was not easy to fully describe all information in the present study at the abstract part. However, it seems that the present abstract contains the objectives (since the effect of heat stress on hypothalamic kisspeptin expression in domestic sows is not yet understood, the present study attempted to identify changes in kisspeptin expression in different seasons), methodology (kisspeptin expression, c-Fos staining, stress-related hormones, reproductive hormones, and number of large ovarian follicles), main findings (decreased reproductive performance, reduced kisspeptin expression/neuronal activation due to high temperature, decreased reproductive hormone levels, and reduced number of large follicles in the ovaries), and conclusion (the present study demonstrated that reduced kisspeptin expression and its neuronal activity in the hypothalamus under heat stress in summer induced downregulation of the HPG axis and caused summer infertility in domestic sows). Therefore, we deeply hope that reviewer #3 favorably reconsiders this comment.
â–ª Comments #5: Keywords - Authors are advised to avoid indexing terms previously reported at the tittle (maybe also in the abstract).
â–ª Response #5: According to ''Instructions for Authors', the animals recommends that the keywords are specific to the article, yet reasonably common within the subject discipline. Therefore, we carefully selected the best keywords that can be specific and representative to our manuscript. During this procedure, using previously indexed terms at the title and abstract was inevitable. Because we obeyed ''Instructions for Authors of animals' for selecting key words, we deeply hope that reviewer #3 favorably reconsiders this suggestion.
â–ª Comments #6: Introduction - At first, authors should correctly write the scientific name of the species Sus scrofa respecting the international rules for this kind of citation. Despite this little mistake, introduction is well written and clearly presents a basis for the study execution. Moreover, objectives are clearly exposed. One point here: despite the interesting design and results, the study is slightly incomplete since it did not present a large evaluation of kisspeptin pattern along all the seasons, and data related to autumn and winter is missing. Anyway, this do not invalidate the study, but remains here as a suggestion for a further evaluation.
â–ª Response #6: We deeply apologize the mistake for the scientific name of the species in the present study. We corrected sus scrofa to 'Sus scrofa' in the manuscript, which can be seen in lane 45. And we highly agree the reviewer`s another comment (kisspeptin pattern in autumn and winter). In fact, now we have designed new study that is to uncover the relationship between kisspeptin and low atmospheric temperature (winter). However, because the present study was mainly focused to reveal the alteration of kisspeptin expression under heat stress during summer (as summer infertility), other factors including low temperature were not considered.
â–ª Comments #7: Methods - I did not understand why the first figure is named as a "supplementary" one; this seems to be unnecessary.
â–ª Response #7: In accordance with 'Instructions for Authors', the animals permit to preparation of supplementary figure to further share research data. Because some readers who are not majored in the anatomy may have difficulties in understanding anatomical terms and morphology of organs, we wanted to prepare some representative images of each organ during samplings. Therefore, if you don`t mind, we hope to maintain this supplementary figure as it is. We deeply hope that reviewer #3 favorably reconsiders this suggestion.
â–ª Comments #8: With regards to the measurement of steroid hormones, was that conducted in an unique moment for the occasion of the slaughtering? Couldn't the slaughter stress influence the hormones profile?
â–ª Response #8: The blood collection was performed from the sows at a mooring in the slaughter (prior to slaughtering). After collecting blood and checking dermal temperature, the animals were than slaughtered. Therefore, we think that the slaughter stress might not influence to hormone profiles in animals. However, we also agree that blood collection step in the present manuscript is seen as it was conducted during slaughtering animals. Therefore, in accordance with reviewer`s comment, we corrected some descriptions which can be seen at lane 114, 135, 138-139, and 143-145.
â–ª Comments #9: Please provide more statistical details. Were the data checked for normality and homocedasticity? Did you need to transform any data? What groups were compared using ANOVA followed of Student's t test? Did you check the existence of correlations among what data? Why did you have to represent data using standard error and deviation, and why not using only one representation for all the data?
â–ª Response #9: Norrmality and homocedasticity of raw data were checked first using SPSS 12.0, thereafter, Student’s t-test or one-way ANOVA with Tukey’s post hoc test was conducted. In case of statistical comparison between 2 groups (SP vs. S), Student’s t-test was involved (Figure 1, Figure 4, and Figure 5E). When significant differences among 4 groups (follicular phase in SP vs. luteal phase in SP vs follicular phase in S vs. luteal phase in S) were statistically assessed, ANOVA with Tukey’s post hoc test was employed (Figure 2, Figure 3, and Figure 5A-5D). During Pearson’s correlation analysis, the different values obtained from all animals (n=50) were compared (dermal and atmospheric temperature; stress hormone level and atmospheric temperature; kisspeptin level and atmospheric temperature; Kisspeptin level and GnRH level). As commented by reviewer #3, our description for the statistical analysis was insufficient. Therefore, relevant corrections were places at lane 257-264 and 290.
â–ª Comments #10: Results - Findings are clear and well presented.
â–ª Response #10: We deeply appreciate to the favorable reviews.
â–ª Comments #19: Discussions - Results are well-discussed. Sometimes, it seems that authors report a lot of literature review, but it generally helps them to explain their results. Moreover, conclusions are adequate and linked to the main aims of the study.
â–ª Response #11: We sincerely appreciate to your positive comment.
â–ª Comments #12: References are ok.
â–ª Response #12: We deeply appreciate to the kind reviews.
.
